# A satellite-based ice fraction record for small water bodies of the Arctic Coastal Plain (2017 to 2023)

Hong Lin[1], Jinyang Du[1], John S. Kimball[1], Xiao Cheng[2], J. Patrick Donnelly[1,3], Jennifer D. Watts[4], Annett Bartsch[5]

[1] Numerical Terradynamic Simulation Group, University of Montana, Missoula MT, USA

[2] School of Geospatial Engineering and Science, Sun Yat-sen University, and Southern Marine Science and Engineering Guangdong Laboratory (Zhuhai), Zhuhai 519082, China

[3] Ducks Unlimited Inc., Missoula MT, USA

[4] Woodwell Climate Research Center, Falmouth, MA, 02540, USA

[5] b.geos, Industriestrasse 1, 2100 Korneuburg, Austria

*Correspondence to*: Jinyang Du (jinyang.du@ntsg.umt.edu) and Xiao Cheng (chengxiao9@mail.sysu.edu.cn)

**Abstract.** Ice cover of water bodies in the northern high latitudes (NHL) is highly sensitive to the changing climate, and its dynamics exert substantial impacts on the NHL ecosystems, hydrological processes, and the carbon cycle. Yet, operational quantification of ice cover dynamics for smaller water bodies (e.g., ≤ 25 km²) over vast, remote NHL regions remains limited. Here, we developed an ice fraction dataset for small water bodies (ponds, lakes, and rivers; 900 m² to 25 km²) across the Arctic Coastal Plain of Alaska (ACP) from 2017 through 2023, using Sentinel-1 Synthetic Aperture Radar (SAR) imagery, texture features, and Daymet air temperature data. The dataset has a spatial resolution of 1 km and a temporal resolution of approximately 6 days. Compared with the Google Dynamic World (DW) product derived from Sentinel-2 optical remote sensing, our dataset shows high consistency with DW ($R = 0.91$, RMSE = 0.19) while having enhanced temporal coverage due to less SAR constraints from solar illumination, cloud cover, and atmospheric conditions. Validation against in-situ observations suggests that our dataset is more capable of capturing small water body ice phenology (e.g., freeze-up and break-up dates) relative to DW, with an 11-day reduction in mean absolute error. Our ice fraction dataset reveals high spatial heterogeneity in ice conditions mainly occurring in June for small water bodies across the ACP. The ice phenology analysis over three selected subregions further shows that a warmer transition period generally leads to earlier ice break-up and later freeze-up, while the responses of ice fraction to warming climate vary among and within individual water bodies. The resulting dataset is anticipated to fill a gap in ice phenology studies for small water bodies, improve our understanding on the interactions between ice dynamics and climate change, and enhance the coupled modelling of ice and carbon processes. The S1 ice fraction dataset is publicly available at https://doi.org/10.5281/zenodo.17033546 (Lin et al., 2025).

## 1 Introduction

Ice cover of rivers and lakes in the northern high latitudes (NHL) is a key indicator of climate change (Adrian et al., 2009). The seasonal dynamics of water ice cover, including freeze-up, break-up, and ice duration, are collectively referred to as ice phenology (Sharma et al., 2020). Changes in ice phenology can exert broad socio-economic and ecological impacts, such as influencing transportation networks (Hori et al., 2018), fisheries resources (Orru et al., 2014), wildlife habitats (Caldwell et al., 2020), hydrological cycle (Wang et al., 2018), and carbon cycle (Matthews et al., 2020). Consequently, lake and river ice are of high scientific relevance and have become an important focus in climate-related research (Culpepper et al., 2024; Yang et al., 2020).

Small water bodies dominate in number among global surface water bodies and contribute significantly to variations in global surface water area and carbon emissions (Mullen et al., 2023; Pi et al., 2022). With the Arctic warming rate exceeding three times the global average (Rantanen et al., 2022), the tens of thousands of Arctic water bodies are experiencing uncertain changes in the extent and timing of seasonal ice cover, which is vital for understanding the arctic carbon, water, and energy cycles (Sharma et al., 2022). Moreover, the rising instability of seasonal ice cover is increasing risks to human welfare in Arctic communities, which depend on frozen lakes and rivers as major conduits for winter travel. In addition, small water bodies are key sources of methane, surpassing big lakes by over tenfold in total emissions due to their high carbon content, low oxygen levels, and shallow nature (Wik et al., 2016). The ice dynamics from these small water bodies thus strongly regulate the magnitude and timing of Arctic methane emissions, which may be increasing and exacerbating global warming (Matthews et al., 2020). Despite the broad importance, knowledge of ice cover dynamics for small water bodies in the vast and remote NHL remains limited partly due to the lack of all-weather satellite observations with high-resolution and frequent-sampling capabilities. One representative study region is the Arctic Coastal Plain of Alaska (ACP), which contains a high density of small surface water bodies (Smith et al., 2007). Since the early 21st century, this region has experienced marked hydrological changes due to climate warming and permafrost thaw (Webb et al., 2022). However, major lake ice observation datasets and related phenological analyses do not include lakes in the ACP (Benson et al., 2000; Sharma et al., 2019, 2022). Studies based on lake modeling also face limitations, as their coarse spatial resolution (e.g., 0.5° or 1°) makes them unsuitable for characterizing ice cover dynamics in small water bodies (Grant et al., 2021; Huang et al., 2022).

Satellite remote sensing is currently the most practical approach for mapping open-water ice over the remote Arctic regions where field measurements and airborne campaigns are very limited. Satellite observations collected using optical-infrared (IR) and active and passive microwave sensors have been widely used for mapping ice cover over large regions (Du et al., 2019).

High-resolution IR satellites such as Planet SuperDove/Skysat, Sentinel-2, Landsat, and Terra/Aqua are particularly useful for delineating ice cover extent from sub-meter to 1000 m scales (Arp et al., 2013; Brown et al., 2022; Mullen et al., 2023; Šmejkalová et al., 2016; Wang et al., 2022; Yang et al., 2020; Zhang et al., 2021). For example, the Google Dynamic World (DW) product characterizes snow and ice conditions along with other land covers globally based on Sentinel-2 observations,

with a revisit frequency of about 4–10 days (Brown et al., 2022). However, the utility of these data is strongly constrained in the Arctic by signal degradation and data loss stemming from extended polar darkness and persistent cloud cover or smoke (Brown et al., 2022). Satellite microwave observations are capable of distinguishing between water and ice due to their contrasting dielectric properties, while exhibiting relatively low sensitivity to solar illumination and atmosphere constraints at lower frequencies ($\sim < 89$ GHz) (Antonova et al., 2016; Du et al., 2017; Kang et al., 2012; Šmejkalová et al., 2016).

Passive microwave radiometers such as the Advanced Microwave Scanning Radiometer–Earth Observing System (AMSR-E/2) provide frequent ($\sim$ daily) but relatively coarse spatial-resolution ($\sim$5–25 km) observations over northern ($\geq 45°$ N) land areas (Du et al., 2017; Kang et al., 2012). For example, a daily lake ice phenology record (5-km resolution) from 2002 to 2021 derived from AMSR-E/2 enabled precise (95 % temporal accuracy) ice cover mapping for Northern Hemisphere lakes regardless of cloud conditions (Du et al., 2017). Despite a general tendency towards thinner ice, later freezing, and earlier break-up in the Northern Hemisphere driven by recent climate warming (Du et al., 2017; Kang et al., 2012; Šmejkalová et al., 2016), the study also revealed opposing trends toward earlier ice formation and later ice break-up existing over specific lakes and periods. However, the coarse resolution of passive microwave sensors restricts their application to only the largest lakes (area $\geq 50$ km$^2$), while similar capabilities for monitoring the abundance of smaller water bodies across the Arctic is lacking.

Space-borne radar instruments are highly sensitive to ice conditions similar to passive microwave sensors, while having comparable resolutions to optical sensors capable of delineating ice cover of small water bodies. The radar open water ice observations are governed by sensor configurations (frequency, polarization, incidence angle) and scattering from or within snow/ice/water/sediment layers (Murfitt and Duguay, 2021). For the ice formation period, the contrasting pattern of high backscatter from cracks and deformations relative to the surrounding thin ice is indicative of the initial ice cover (Antonova et al., 2016). As the ice thickness grows, backscatter generally increases due to the roughness and large dielectric contrast at the ice and water surface (Murfitt and Duguay, 2021). When maximum ice thickness is reached, decreased radar backscatter can be observed from bedfast ice due to the small dielectric contrast at the ice and ground surface. For the melting period, dark patches/spots in radar images may be observed from open-water areas or small water pools on ice; while increased backscattering is also expected from the roughened ice surface during melt and refreeze events (Murfitt and Duguay, 2021). Accordingly, statistics-based approaches have been widely used to distinguish ice and water (Engram et al., 2018; Murfitt and Duguay, 2021). Machine learning approaches were recently utilized to leverage the characteristic radar backscatter patterns observed at different ice freezing/thawing phases for enhanced ice cover detection (Tom et al., 2020). Despite the algorithm development, radar capabilities for routine lake ice monitoring over large regions have been constrained by the complex interactions between microwave and water body features, limited global coverage, and relatively sparse temporal frequency of sampling from prevailing satellites (Du et al., 2019). There remains a lack of databases that can provide all-weather and operational observations of ice cover and phenology dynamics for small water bodies across the ACP, where accelerated warming and thawing occurs.

In this study, we developed a dataset of ice fraction for smaller water bodies (≤ 25 km²) on the ACP using Sentinel-1 SAR data (S1), with a temporal resolution of about 6 days. The total area of the studied water bodies is 6,443.59 km², with the smallest unit measuring 900 m². We analyzed 3,717 S1 images acquired between 2017 and 2023. A random forest (RF) classifier was applied to each image to generate 10-m resolution ice cover maps. The study area was subsequently divided into 1 km² grid cells, and the ice fraction of small water bodies within each grid was calculated. The reliability of the dataset was evaluated based on classification accuracy, comparison with DW, and validation against observed ice phenology data. We also applied the resulting dataset to quantify multi-year ice fraction patterns during the melting season across the ACP, and assessed the potential utility of the data record for monitoring the regional ice phenology.

## 2 Study area and data set

### 2.1 Study area

Our study focused on ice cover conditions of small water bodies (900 m² to 25 km²) across the ACP (Fig. 1a). We selected three representative regions (Fig. 1b–d) and estimated the ice phenology of small water bodies within each region based on satellite-derived ice fraction data (see Section 3.6). These regions differ in latitude, longitude, and geomorphological characteristics. Region 1 (Fig. 1b) lies near the northern coast of the ACP, adjacent to the Barrow flux tower, and features thermokarst lakes (Arp et al., 2012). Region 2 (Fig. 1c) is underlain by an ancient sand dune field and contains relatively deep lakes (Simpson et al., 2021). Region 3 (Fig. 1d) is situated near Prudhoe Bay, which is the largest conventional oil field in North America (Jamison et al., 1980) and is characterized by extensive infrastructure that supports oil and gas exploration. The water bodies in Region 3 provide freshwater resources for local industrial activities and are subject to greater human disturbance.

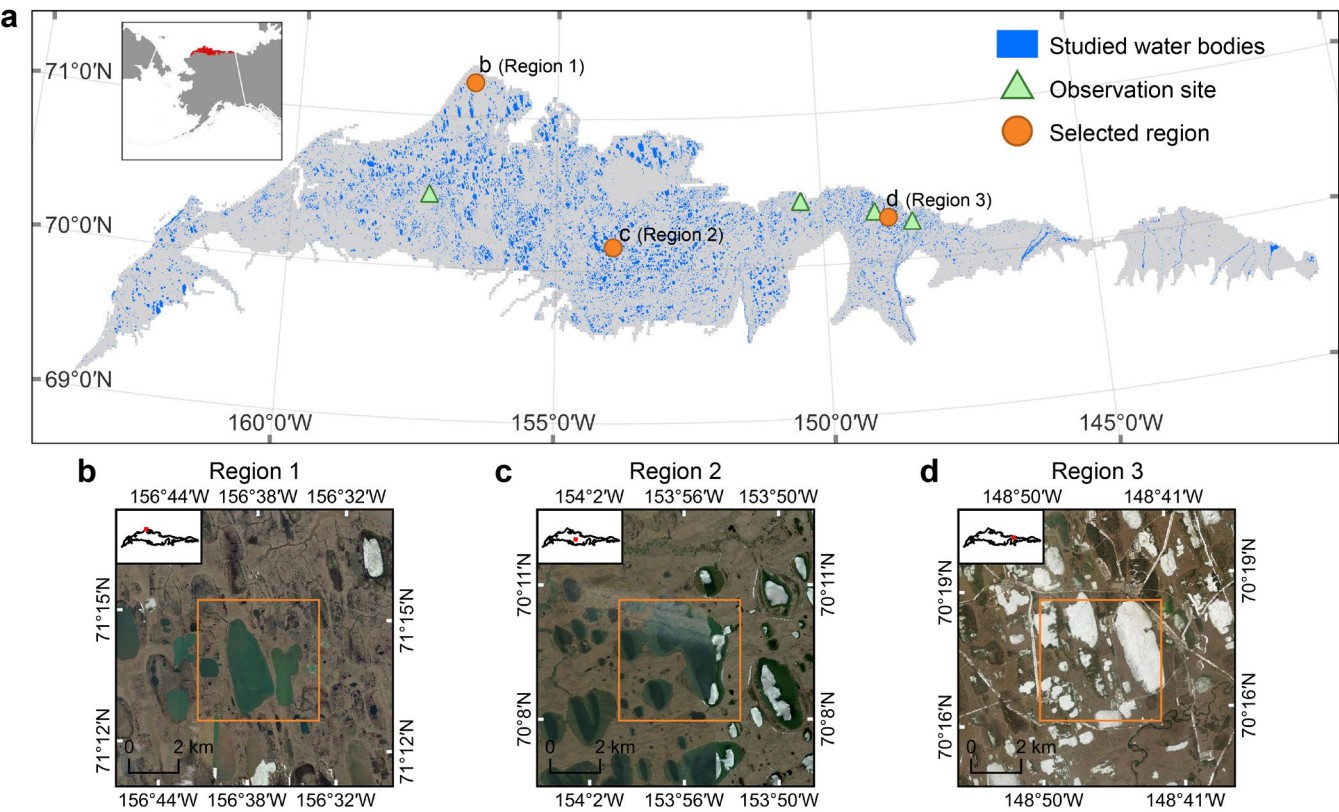


**Figure 1.** Numerous small water bodies are distributed across the study area. a, Small water bodies (blue) investigated within the ACP,
locations of observed ice phenology records (green triangles), and the three selected regions for ice phenology analysis (orange circles). b–
d, Enlarged views of the three selected regions, with orange borders indicating 5 × 5 km areas. Panels b–d use basemaps from Esri World
Imagery.
**2.2 Data set**
**2.2.1 Water body data**
We used the Global Lakes and Wetlands Database Version 2 (GLWD v2) (Lehner et al., 2024) and the Joint Research
Centre Global Surface Water (GSW) products, including maximum water extent and water occurrence (Pekel et al., 2016), to
generate the small water body mask (Fig. 1a, Section 3.1.1). GLWD v2 provides a global map of inland surface water with a
spatial resolution of 15 arc-seconds, including 33 categories of water bodies such as lakes (Lehner et al., 2024). The GSW
maximum water extent product provides the maximum extent of surface water between 1984 and 2021 at 30-m spatial
resolution, while the GSW water occurrence product provides the frequency of surface water presence at each pixel during
1984–2021 (Pekel et al., 2016).

### 2.2.2 Sentinel-1 SAR data

In this study, we used Sentinel-1 SAR imagery to detect ice within small water bodies in the ACP. The Sentinel-1 constellation includes two satellites, S1A and S1B, which were launched in 2014 and 2016, respectively, with sun-synchronous descending/ascending orbits and 6 AM/PM mean local sampling times. The revisit frequency of the Sentinel-1 constellation is 6 days from both satellites but was reduced to 12 days after the S1B satellite ceased operating in December 2021 (https://sentinels.copernicus.eu/copernicus/sentinel-1). SAR is less affected by cloud cover or illumination conditions relative to optical-IR sensors, enabling all-weather monitoring. Due to differences in dielectric properties and surface roughness, ice and water typically exhibit distinct backscatter characteristics in SAR imagery (Stonevicius et al., 2022; Section 1) for facilitating ice/water classifications. The Sentinel-1 Ground Range Detected (GRD) product, which was generated from the SAR observations under Interferometric Wide Swath (IW) mode and both ascending and descending orbits, was used in the study. The vertically (VV) and cross (VH) polarized radar backscatters and their incidence angles were analysed for ice and water mapping. A total of 3,717 S1 images (1,451 ascending and 2,266 descending scenes) covering the ACP at 10-m resolution were collected for the period from 2017 through 2023.

### 2.2.3 Dynamic World and Sentinel-2 data

Dynamic World (DW) is a 10-m resolution land cover dataset derived from Sentinel-2 (S2) optical-IR imagery. It includes nine classes, with snow and ice among them (Brown et al., 2022). Due to the application of cloud filtering, the temporal resolution of the DW product is approximately half that of Sentinel-2, around 4–10 days (Brown et al., 2022). The land cover classes in DW are predicted using a fully convolutional neural network (FCNN), with the snow/ice class achieving a user accuracy of 71.2 % and a producer accuracy of 94.2 % (Brown et al., 2022). The DW data were used for training the RF model and validating S1-based classifications. Specifically, we used the label band from DW, which represents the land cover class label with the highest estimated probability. Then, the DW images from 2017 through 2023 were converted into binary ice and non-ice masks (i.e., label dataset) for the identified small water bodies in the ACP. DW is generated from S2 images with cloud cover ≤ 35 %. However, to ensure higher-quality samples for training the RF model, we paired DW scenes with S2 imagery acquired on the same day and retained only those with cloud cover ≤ 20 %.

### 2.2.4 Daymet data

The Daily Surface Weather and Climatological Summaries (Daymet V4) dataset provides daily estimates of surface weather parameters over North America at 1-km spatial resolution since 1980 (Thornton et al., 2021). The air temperature parameters of Daymet V4 are estimated through a weighted multivariate regression model based on observed weather station data. The cross-validation results of Daymet V4 show that the average daily mean absolute error (MAE) is 1.78 °C for daily minimum air temperature (Tmin) and 1.52 °C for daily maximum air temperature (Tmax) (Thornton et al., 2021). For each image from 2017 through 2023, we selected Tmax, Tmin, and a 5-day lagged mean air temperature (Tmean5d) as part of the

RF predictors. To derive the Tmean5d, we first computed the daily average air temperature (Tavg) by averaging Tmax and
Tmin, then calculated the mean Tavg over the current day and the preceding four days. Temperature was selected as a predictor
because it is a key atmospheric factor influencing the dynamics of ice cover (Woolway et al., 2020). Considering the data
uncertainties due to the spatial interpolation from limited station measurements over the region, we added random noise of
±1.5 °C to the original Daymet Tmax, Tmin, and Tmean5d data to improve the robustness of the RF model.

### 2.2.5 Observed ice phenology data

To assess the accuracy of our ice fraction dataset in estimating ice phenology, we collected seven observational records
from four rivers within the study area (Fig. 1a) from the River and Lake Ice Phenology Dataset for Alaska and Northwest
Canada (Arp and Cherry, 2022). The seven collected records are derived from ground-based visual observations. This
observational dataset provides information such as the dates of ice break-up and freeze-up, and the coordinates of each record
(Table S1). The dataset does not provide specific definitions for break-up and freeze-up dates, nor does it include specific
accuracy metrics, but it notes that observations of river and lake ice conditions are primarily conducted by shore-side
community members and are qualitative in nature.

## 3 Methods

## 3.1 Data preprocessing

### 3.1.1 Generating small water body mask

To delineate the extent of small water bodies, we first extracted lakes larger than 25 km² from the GLWD v2 product.
These large lakes were then removed from the GSW maximum water extent to generate an initial mask of small water bodies.
To reduce ice detection errors along littoral zones and riverbanks, we used the GSW water occurrence product to retain only
areas with water occurrence greater than 80 % within the initial small water body extent. The total area of the remaining small
water bodies used in this study is 6,443.59 km². We applied the resulting water body mask to the S1 imagery, the DW product,
and the Daymet air temperature data for ice cover mapping over small water bodies in the ACP.

### 3.1.2 Pre-processing of Sentinel-1 imagery

The S1 data were pre-processed before being fed into the RF model, which included incidence angle normalization, de-
speckling using Refined Lee filtering, texture calculation, image clipping, and water body masking (Fig. 2a).
To correct for SAR incidence angle effects (Koyama et al., 2019), we normalized the incidence angles to 40° using a well-
established cosine correction method (Mladenova et al., 2012) (Eq. 1).
$\sigma_{corrected} = \sigma_\theta \left(\frac{cos\ \theta_{ref}}{cos\ \theta}\right)^n$     (1)
$$ln(\sigma_\theta) = n \times ln(cos\ \theta) + b \tag{2}$$
where $\sigma_\theta$ is the backscatter coefficient of a pixel in the SAR image, $cos\ \theta$ is the cosine of the incidence angle for that pixel,
and $cos\ \theta_{ref}$ is the cosine of the reference incidence angle (set to 40° in this study). $\sigma_{corrected}$ represents the backscatter
coefficient corrected to the reference angle. The exponent $n$ describes surface roughness, and $b$ is the intercept of the linear
equation. The exponent $n$ in Equation (1) is derived by performing a linear fit between $ln(\sigma_\theta)$ and $ln(cos\ \theta)$, as shown in
Equation (2).
Based on Equation (1), we applied incidence angle normalization to the VV and VH bands of both ascending and
descending S1 data. To determine the four corresponding $n$ values, we sampled each S1 image within the small water body
extent during the study period. A total of 318,400 data points were collected from ascending orbit images and 401,560 data
points from descending orbit images. The correction coefficients ($n$) derived using Equation (2) were 4.43 and 2.6 for
ascending VV and VH, and 4.11 and 2.6 for descending VV and VH, respectively.
To reduce SAR speckles, we applied the Refined Lee filter (Lee and Pottier, 2017) to each S1 image after incidence angle
normalization. We then calculated the correlation texture for the VV band (VV_corr), which quantifies the similarity between
a pixel and its neighbors. The inclusion of radar backscatter texture information provides spatial context for the RF model,
enabling the classifier to utilize not only the value of individual pixels but also statistical characteristics of their surrounding
neighborhood. For example, VV_corr texture information is indicative of ice and water conditions (e.g., ice patches, open
water patches, and ice-water boundaries) during the break-up period (Fig. S1), and thus supports the machine-learning based
classification. Among the commonly used texture features (Soh and Tsatsoulis, 1999), such as correlation, variance, contrast,
energy, and entropy, VV_corr was found to be most important for ice detection. Considering the importance of VV_corr and
the increased computational burden introduced by multiple texture features, we ultimately selected VV_corr as the only texture
predictor.
To avoid the impacts of S1 degradations over image edges, we removed the pixels within a 100-m buffer area from the
image edges. Finally, the small water body mask was applied to each S1 scene.

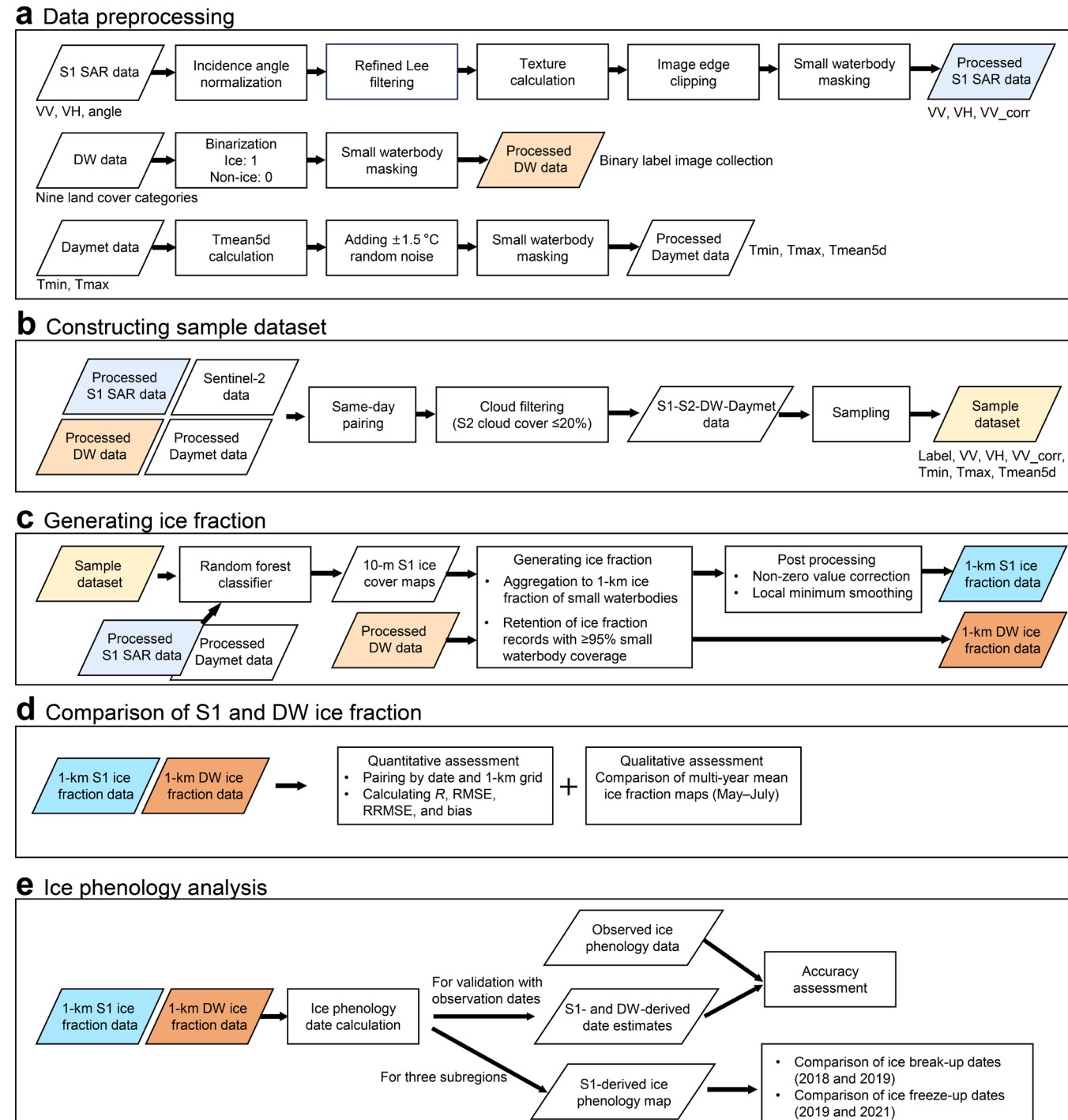

**Figure 2.** Workflow for generating ice fraction dataset, comparing S1 and DW ice fraction, and analyzing ice phenology.

## 3.2 Constructing sample dataset

To train and test our RF-based ice detection model, we first constructed a dataset using DW ice/non-ice classifications as predictand, and S1 VV and VH backscatter coefficients, VV_corr, and Daymet air temperature variables as predictors (Fig. 2b). We paired all same-day S1, S2, DW, and Daymet images over the study area from 2017–2023 to form the S1-S2-DW-Daymet image collection. The ACP study area was divided into 23 longitudinal zones at 1° intervals. For each S1-S2-DW-Daymet image pair within a given longitudinal zone, we sampled 20 points per class (ice and non-ice) based on the DW labels.

In total, we collected 51,858 samples, consisting of 31,579 ice samples and 20,279 non-ice samples (Fig. 3). Of these, 19,124 samples (11,466 ice and 7,658 non-ice) were from ascending orbit scenes, and 32,734 samples (20,113 ice and 12,621 non-ice) from descending orbit scenes. We separated the ascending and descending orbit samples to develop independent ice detection models for each orbit.

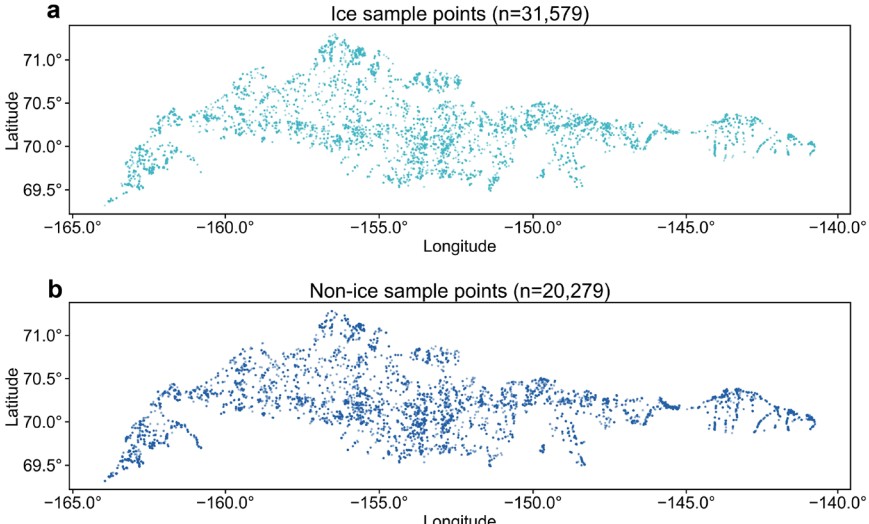

**Figure 3.** The labelled data used for training and testing the ice detection models. a, Spatial distribution of ice samples. b, Spatial distribution of non-ice samples.

## 3.3 Generating ice fraction

We used the RF model to detect ice cover in small water bodies, specifically performing binary classification (ice vs. non-ice) at the pixel scale (Fig. 2c). The RF is an ensemble learning algorithm with high computational efficiency and robustness against overfitting (Belgiu and Drăguţ, 2016; Maxwell et al., 2018). The input features for the ice detection model included six predictors: VV, VH, VV_corr, Tmax, Tmin, and Tmean5d. The predictand is the ice/non-ice classification. Considering the different passing time, S1 ascending (6 PM local time) and descending (6 AM local time) observations were processed separately. Therefore, separate RF models were trained using S1 ascending and descending orbit samples. The dataset was randomly split into 80 % for training and 20 % for testing (Table S2). Optimal model hyperparameters were determined through

tuning (Table S3). Model performance was evaluated using the test set, with metrics including overall accuracy, user accuracy,
and producer accuracy.
The trained ice detection models were applied to each S1 image throughout the study period to produce 10-m resolution
ice cover maps for small water bodies in the ACP (Fig. 2c). The study area was then divided into 1-km grid cells, and the ice
fraction was calculated for each grid based on the 10-m ice cover maps. Only grid cells with at least 95 % spatial coverage of
small water bodies with RF ice classifications were retained. The 1-km ice fraction dataset from 2017–2023 was generated by
merging the results for the respective S1 ascending and descending observations. In addition, we generated a corresponding
1-km ice fraction dataset from the DW product for inter-comparisons.
**3.4 Post processing**
Post-processing was performed to minimize retrieval uncertainties and remove outliers from the ice fraction record. We
first corrected non-zero ice fraction values during the ice-free season, which are likely artifacts from the misclassifications
under rough water surface conditions (Du et al., 2016). Specifically, the 1-km S1 ice fraction time series for each grid was
divided into one-year segments, and a Gaussian smoothing filter was applied to each segment. Periods with smoothed ice
fraction values below 0.5 were identified as water-dominated periods. Within these periods, we located the first and last days
in the original (i.e., unsmoothed) series where the ice fraction dropped below 0.1 and set all values between the two dates to
zero.
During the ice break-up process, the S1 radar backscatter tends to be reduced first due to an increase in liquid water content
in snow on top of ice cover, followed by a possible increase with greater snow and ice surface roughness as melting continues
(Murfitt et al., 2024). In the ice fraction time series, this effect can manifest as a dip followed by a rise. To address the snow
melting impacts, we further applied a local minimum smoothing filter to the ice fraction time series. Specifically, if a given
value was lower than both its preceding and following values, it was replaced by the average of those two neighbouring values.
**3.5 Comparison of S1 and DW ice fraction**
In addition to the RF model evaluations (Sections 3.3 and 4.1), we also compared the S1 and DW ice fraction datasets
(Fig. 2d). For quantitative comparison (Section 4.2), we paired each S1 ice fraction value with the corresponding DW value
on the same day and grid during 2017–2023. Only grids where small water bodies cover at least 5 % of the grid area were
included. We then calculated the Pearson correlation coefficient ($R$), root mean square error (RMSE), relative RMSE (RRMSE),
and bias between S1 and DW ice fraction datasets. For qualitative comparison (Section 4.3), we used S1 and DW ice fraction
datasets to generate maps of the multi-year average ice fraction for small water bodies in the ACP during May to July and
analyzed their spatiotemporal patterns.

## 3.6 Ice phenology analysis

The ice fraction dataset captures the small water body ice phenology, such as the freeze-up and break-up dates. Freeze-up refers to the process from the initial formation of ice to full ice coverage, while break-up refers to the transition from the initial fracturing of ice to the return of open water conditions (Sharma et al., 2022). Definitions of freeze-up and break-up dates vary across studies, commonly based on the initiation or completion of these phases (Arp et al., 2013; Brown and Duguay, 2010; Sun, 2018). In this study, we define the break-up date as the first day on which the ice fraction drops below 0.95, and the freeze-up date as the last day below 0.95 before the ice fraction rises above this threshold. The 0.95 threshold represents the onset of break-up and the completion of freeze-up.

To evaluate the accuracy of remote sensing-based ice fraction data in estimating ice phenology, we extracted S1 and DW ice fraction values from the 1-km grid cells corresponding to the in-situ ice phenology records (Fig. 1a, Table S1). Subsequently, ice phenology dates based on S1 and DW were estimated and compared with the observed dates (Fig. 2e). We also estimated ice phenology for three 5 × 5 km regions (Fig. 1b–d) within the ACP representing distinctive lake geomorphological characteristics and compared the results between warm and cold years for understanding the impacts of changing climate on lake ice dynamics (Fig. 2e).

## 3.7 Uncertainty assessment

To assess the S1 ice fraction data uncertainty, we collected all paired DW and S1 ice fraction observations on the same dates from 2017 through 2023 for each grid cell. Subsequently, the RMSE of S1 and DW ice fractions was calculated for each grid cell and normalized by the average DW ice fraction of that grid cell across all temporally matched observations, to derive the RRMSE, expressed in percentage. The RRMSE for each grid cell is calculated as follows:

$$RRMSE = \frac{\sqrt{\frac{1}{n}\sum_{i=1}^{n}(S1_i - DW_i)^2}}{\overline{DW}} \times 100\% \tag{3}$$

where $S1_i$ and $DW_i$ denote the S1 and DW ice fraction of the same 1 km grid cell at the $i$-th temporally matched observation, respectively; $n$ is the number of temporally matched observations available for that grid cell; and $\overline{DW}$ represents the mean DW ice fraction of that grid cell across all $n$ temporally matched observations. The resulting RRMSE map serves as a quality flag layer for the ice fraction product and will be released alongside the final dataset (Fig. S2, Section 6).

## 4 Results

### 4.1 Performance of 10-m ice detection

Our RF results were highly consistent with the DW for pixel-based ice classifications. For S1 ascending orbits, the overall accuracy, user accuracy, and producer accuracy were 0.91, 0.93, and 0.92, respectively. For descending orbits, these metrics were 0.91, 0.92, and 0.93. Both temperature-based and radar-based features are important, with comparable contributions to the predictions (Fig. S3). The temperature variables provide regional temperature distributions and seasonal context, whereas the SAR variables provide the direct observations crucial for distinguishing pixel-level ice conditions. Figure 4 shows good consistency between the 10-m S1 ice cover maps and the DW ice cover maps on the same days. Unlike the DW ice cover maps, which suffer from large data gaps due to cloud contamination, the S1 imagery provides valid estimates of ice cover under cloudy conditions (Fig. 4b). Due to differences in overpass times on the same date (UTC 03:00 for ascending S1, UTC 17:00 for descending S1, and UTC 22:00 for S2), S1 and S2 may capture different ice conditions when active ice melting occurred (e.g., Fig. 4a).

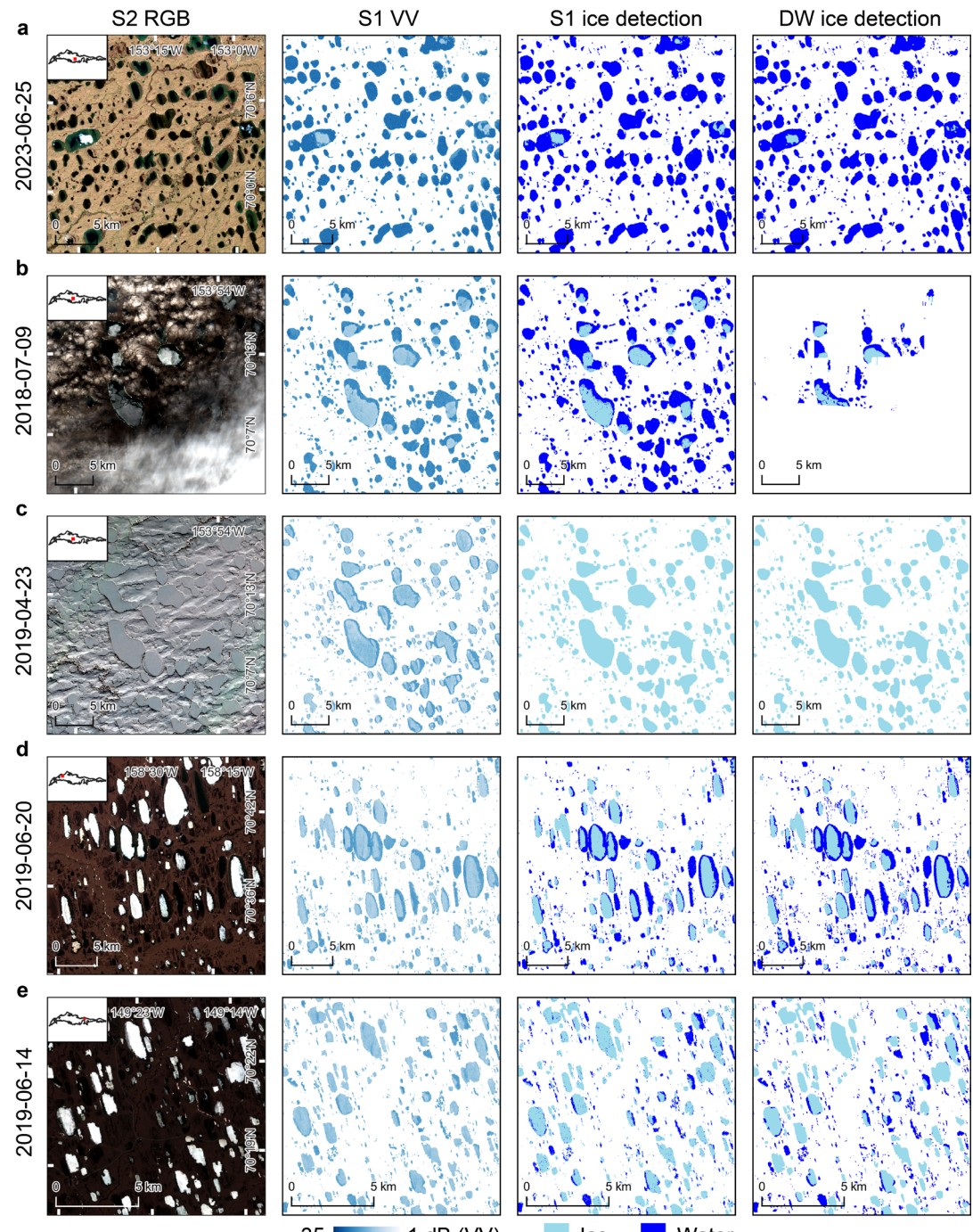

297

**Figure 4.** Comparison of Sentinel-2 RGB, Sentinel-1 VV, S1-based ice detection, and DW-based ice detection. Rows a–e show results from different periods and regions. Each row of subplots presents, for the same day, the optical image, SAR image, the 10-m S1 ice cover map from this study, and the 10-m ice cover map from DW. In row b, significant portions of the S2 optical image and associated DW ice classification are degraded by cloud contamination, whereas the S1 SAR based ice classification is unaffected by this atmosphere constraint.

## 4.2 Quantitative assessment of 1-km ice fraction

Quantitative comparisons showed high consistency between the 1-km S1 and DW ice fraction results, with an $R$ of 0.91, RMSE of 0.19, RRMSE of 28.41 %, and a bias of 0.02 (Fig. 5). The S1 ice fraction shows a slight overestimation relative to DW in low-value areas, such as when the ice fraction ranges from 0 to 0.4. However, this overestimation accounts for less than 10 % of the validation data pairs. Annual comparisons from 2017 through 2023 also showed good consistency, with $R$ values ranging from 0.86 to 0.94, RMSE between 0.13 and 0.22, RRMSE between 0.16 and 0.38, and bias ranging from –0.02 to 0.05 (Table S4).

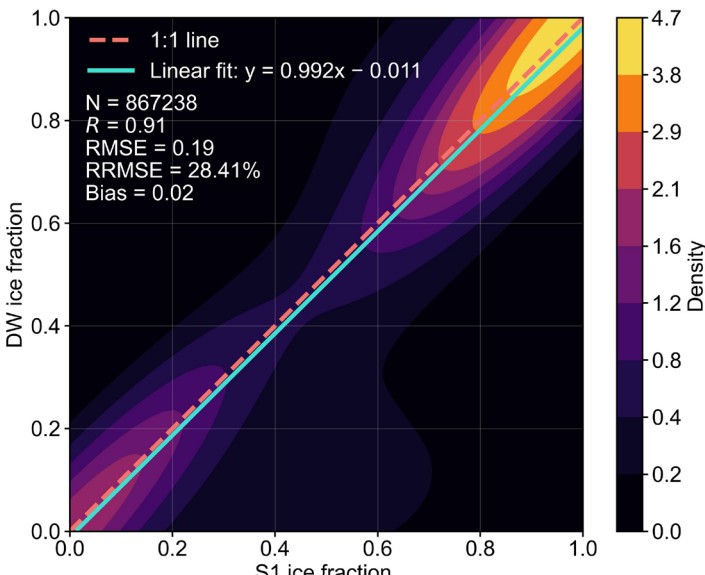

**Figure 5.** Comparison of 1-km S1 and DW lake ice fraction results on the same days from 2017 through 2023 show good agreement. The linear regression line and evaluation metrics ($R$, RMSE, RRMSE, and bias), based on 867,238 data points, are shown in the plot. The background is a kernel density estimate generated through random sampling.

The uncertainty of the S1 ice fraction dataset for each 1 km grid cell was evaluated using RRMSE. About 5.16% of the 1 km grid cells have an RRMSE below 10%, indicating that in these areas the model achieves high accuracy, with predictions closely matching the DW ice fraction (Table 1). An additional 17.30% of the data fall within the 10–20% range, reflecting good model performance (Table 1). The largest portion of the dataset, 31.17%, lies in the 20–30% range, indicating moderate accuracy for a substantial part of the ice fraction predictions (Table 1). Furthermore, 21.27% of the data are within the 30–40% range, showing areas with larger errors. In addition, 11.31% of the data have RRMSE values between 40–50%, and 13.79% exceed 50%, highlighting regions where the model performs poorly. For these high-error areas, users should exercise caution and can filter them using the provided quality layer. Relatively high errors were mainly found along rivers as well as in very small water bodies, where mixed land and water/ice conditions are likely found in S1 observations (Fig. S2). For example,

among the 1-km grid cells with RRMSE greater than 60%, 46.7% areas are distributed in river areas determined by a 1 km buffer around the river centerlines. Contaminations in S1 observations from the surrounding land areas of the elongated or very small water bodies likely led to the large classification uncertainties.

**Table 1.** Uncertainty distribution of S1 ice fraction data. The uncertainty for each 1 km grid cell is measured by the RRMSE of ice fraction between S1 and DW for that cell.

| Uncertainty range | Proportion of 1-km grid cells |
|---|---|
| < 10 % | 5.16 % |
| [10 %, 20 %) | 17.30 % |
| [20 %, 30 %) | 31.17 % |
| [30 %, 40 %) | 21.27 % |
| [40 %, 50 %) | 11.31 % |
| > 50 % | 13.79 % |

## 4.3 Spatiotemporal patterns of ice fraction

We calculated monthly mean lake ice fraction maps for the ACP in May, June, and July of each year from 2017 through 2023, and then averaged these to produce multi-year mean ice fraction maps for each month (Fig. 6). The May composite shows widespread ice coverage over small water bodies across the ACP (Fig. 6a). June marks a period of rapid melt, with a general decrease in ice fraction from higher to lower latitudes. Most rivers exhibit melt conditions in June, and adjacent lakes also show reduced ice coverage during this period (Fig. 6b). This pattern is related to the spring flood pulse and delivery of snowmelt runoff by river inflows from surrounding lake–watershed systems (Brown and Duguay, 2010). By July, ice fractions are minimal, which suggests most small water bodies have completed ice break-up (Fig. 6c). This is consistent with previous findings indicating that ice-out dates for lakes in the ACP generally occur after the summer solstice (Arp et al., 2013).

The multi-year mean ice fraction maps derived from S1 (Fig. 6a–c) and DW (Fig. 6d–f) show good agreement, especially for May and July. In June, when ice melt is most dynamic, the DW ice fraction map indicates higher ice fractions in the western to central ACP, but lower ice fractions in the northeastern-central region compared to the S1 ice fraction map (Fig. 6b,e). These differences are attributable to differences in observation timing between the two datasets. In the western to central ACP, DW observations are concentrated in early June (Fig. S4b), generally capturing pre-melt conditions, whereas the S1 observations in this region are more concentrated in mid-June (Fig. S4a), reflecting more advanced melt. In contrast, in the northeastern-central ACP, DW observations are concentrated in late June (Fig. S4b) when ice had largely melted, while the S1 observations occurred earlier in the month (Fig. S4a) when ice was still present.

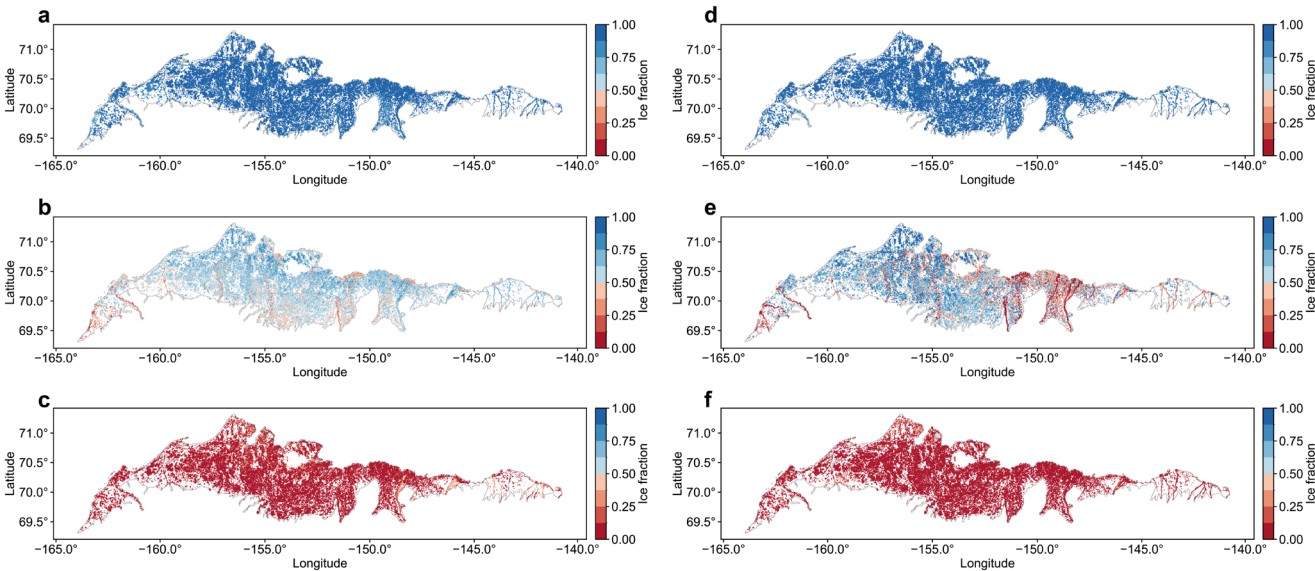

347

**Figure 6.** Multi-year mean 1-km ice fraction maps for small water bodies in the ACP from May to July. a–c show the S1-derived ice fraction
results for May (a), June (b), and July (c). d–f show the DW-derived ice fraction results for May (d), June (e), and July (f). Only grid cells
where small water bodies cover ≥ 1 % of the area are shown.

## 4.4 Ice phenology assessment

For ice phenology estimation, the S1 derived ice fraction record produced more accurate results, with an overall MAE of
7 days, whereas the DW-derived estimates had an overall MAE of 18 days in relation to the 7 ice phenology observations from
the 4 ACP sites (Table 2). The error range for phenology dates derived from the S1 ice fraction record was 0 to 19 days (Table
S1), comparable to a previous study at Lake Hazen in Canada using Sentinel-1 imagery (2–17 days) (Murfitt and Duguay,
2020). In contrast, DW-derived phenology dates showed larger errors, ranging from 5 to 38 days. The estimation error of
freeze-up dates from the S1 ice fraction record is larger than that of break-up dates, mainly due to the record from 2017 (Table
S1). Notably, both S1 and DW estimates show large errors for this period. The S1-based ice fraction data captured the ice
phenology within 1-km grid cells, whereas the in-situ data set were from eye-based visual observations. Therefore, the two
phenology measurements may differ due to mismatches in spatial extent and time of observation.


**Table 2.** Errors in ice phenology dates estimated from 1-km S1 and DW ice fraction data. Errors are presented as mean absolute error (MAE),
calculated based on the results in Table S1. The table includes MAE for break-up dates, freeze-up dates, and overall MAE.

| Data | MAE (days) | | |
|---|---|---|---|
| | Break-up | Freeze-up | Overall |
| DW | 18 | 19 | 18 |
| S1 | 4 | 13 | 7 |


The ice phenology results derived from the S1 ice fraction record captured the impact of an anomalous heatwave in 2019
(Fig. S5a), which led to notably earlier break-up dates across all three regions compared to 2018 (Fig. 7). In June 2019, Region
2 experienced the highest mean air temperature among the three regions, reaching approximately 6 °C (Fig. S5a).
Correspondingly, Region 2 also exhibited the earliest break-up dates among the three regions (Fig. 7d–f). In 2018, some
irregularly shaped and smaller water bodies in Region 3 experienced earlier ice break-up (Fig. 7c). This pattern is consistent
with previous findings suggesting that lakes with more complex shapes and smaller areas tend to break up earlier (Arp et al.,
2013). Our results also showed that freeze-up in 2021 occurred significantly earlier than in 2019 across all three regions (Fig.
8), which was related to a colder September in 2021 (Fig. S5b).

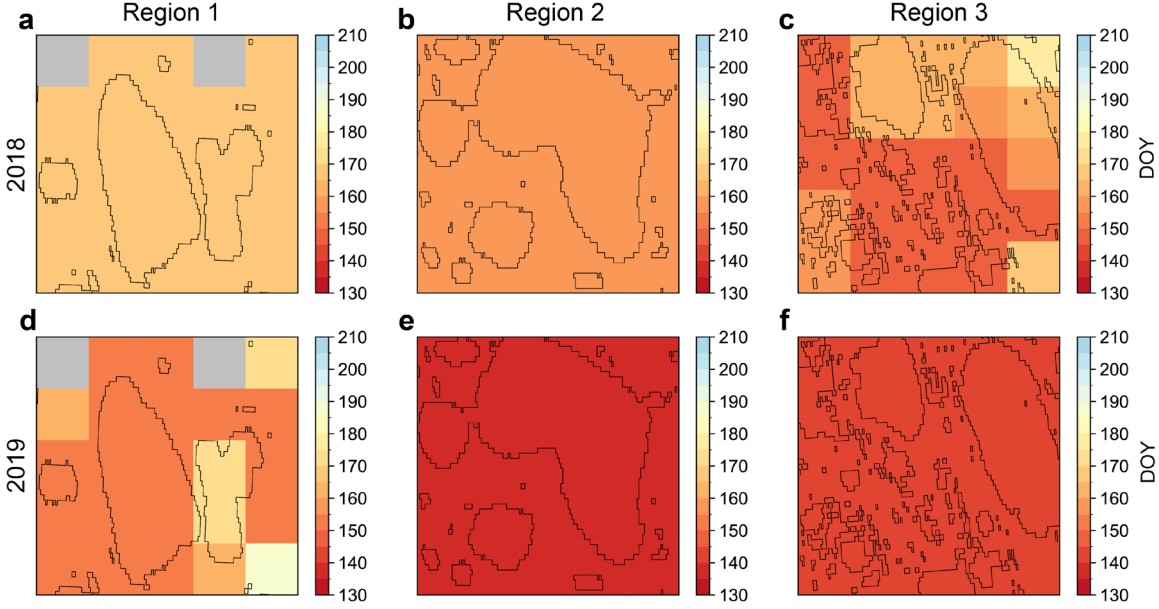


**Figure 7.** The lake ice break-up dates in 2019 (d–f) were generally earlier than those in 2018 (a–c) across the three selected regions in the
ACP (Fig. 1). Each subplot shows a 5 × 5 km area where water bodies are delineated by black lines. The phenology dates are calculated
based on 1-km S1 ice fraction data, and the color of each 1-km grid cell indicates the break-up date in day of year (DOY).

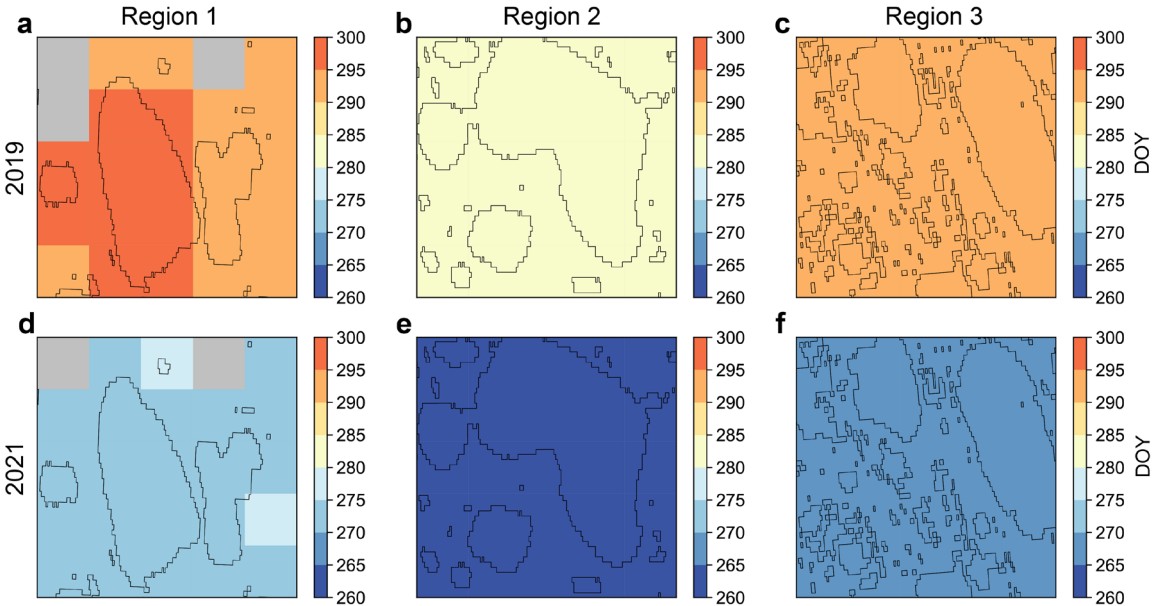


**Figure 8.** The lake ice freeze-up dates in 2021 (d–f) were earlier than those in 2019 (a–c) across the three selected regions in the ACP (Fig.
1). Each subplot shows a 5 × 5 km area with water bodies delineated by black lines. The phenology dates are calculated based on 1-km S1
ice fraction data, and the color of each 1-km grid cell indicates the freeze-up date in day of year (DOY).
**5 Discussion**

This study provides a 1-km resolution ice fraction dataset for small water bodies in the ACP from 2017 through 2023. The

dataset is derived from a 10-m resolution S1 derived ice classification and includes water bodies as small as 900 m² in size,
which have been largely omitted in most previous remote sensing-based studies (Arp et al., 2013; Du et al., 2017; Šmejkalová
et al., 2016; Wang et al., 2022; Zhang et al., 2021). By leveraging the all-weather and day-night observation capability of
satellite SAR sensors, the dataset provides more timely ice cover detection relative to the optical-IR observations. In particular,
optical-IR imagery may fail to reliably capture freezing events in high-latitude lakes due to low solar elevation angles
(Šmejkalová et al., 2016), or miss critical ice information due to cloud contamination. Compared to the operational DW
classifications, our dataset offers higher accuracy in ice phenology estimation (S1 MAE = 7 days; DW MAE = 18 days) and
is more capable of capturing ice dynamics during periods of rapid change (Table 2). This dataset provides a new resource for
tracking small water body ice dynamics complementary to optical-IR results and contributes to enhanced monitoring of NHL
environmental changes.

Our dataset shows that small water bodies within ACP are generally covered by ice in May, experience major melting

events in June, and become mostly ice-free in July (Fig. 6a–c). This pattern is consistent with the DW-based results (Fig. 6d–
f) and previous studies showing ACP ice break-up events mainly occurring between June and July and ending after the summer
solstice (Arp et al., 2013). Our ice fraction maps also show substantial spatial heterogeneity in June ice cover, including a

latitudinal gradient of decreasing ice cover toward the south and lower ice fractions in rivers and nearby water bodies (Fig. 6b). The ice cover conditions of water bodies may vary with their distances from and connectivity with rivers due to the influence of snowmelt runoff on river–lake systems (Brown and Duguay, 2010; Prowse et al., 2011; Woo and Heron, 1989). Compared to isolated lakes, those connected to rivers tend to break up earlier due to the inflow of relatively warmer meltwater (Arp et al., 2013).

Ice phenology analysis suggests that air temperature is the main control of break-up and freeze-up events of small water bodies in the ACP (Figs. 7 and 8), which was also confirmed from satellite observations over large lakes (Du et al., 2017) and in-situ observations over lakes and rivers (Weyhenmeyer et al., 2011). In addition, our results show that break-up dates tend to occur earlier in smaller and more irregularly shaped water bodies (Fig. 7c), which is consistent with previous findings (Arp et al., 2013).

Our ice fraction retrievals are highly correlated with the DW results ($R = 0.91$). The uncertainties mainly arise from the complexity of SAR observations of ice and water, as well as the limitations in the RF training data and ancillary inputs. For example, wind-induced surface roughness may cause strong radar backscatters from open water, leading to water misclassified as ice (Du et al., 2016). On the other hand, the decrease in backscatters caused by melting snow may lead to ice misclassified as water (Murfitt et al., 2024). In addition, thin ice with a smooth surface may appear dark in S1 backscatter images, leading to ice misclassified as water (Mahmud et al., 2022). As a result, wind effects may lead to nonzero ice fraction values during ice-free periods, while wet snow and thin ice can cause anomalous drops in ice fraction. Moreover, lake water salinity can affect SAR backscatter coefficients and thus influence ice detection (Engram et al., 2018). This aspect warrants further investigation in the future. Our ice detection models were trained with diverse samples, yet occasional misclassifications remain unavoidable. Through post-processing, these residual effects were effectively reduced by identifying and removing outliers (Section 3.4). Errors can also arise from uncertainties in the RF temperature predictors. Due to the relatively coarse spatial resolution and reliance on sparse in-situ observations of the Daymet product, the ice detection performance may be affected by the zonal patterns in temperature. By introducing random noises in the Daymet temperatures (Section 2.2.4), this issue was effectively mitigated due to decreased RF sensitivity to temperature. In addition, the RF model training was constrained by the limited availability of a valid DW product due to frequent unfavorable conditions for S2 optical-IR remote sensing in the ACP. For example, periods without S2 clear-sky observations were under-represented in the RF training.

The error in ice phenology estimation based on the S1 ice fraction dataset (MAE = 7 days) is close to the dataset's temporal resolution (~6 days). The uncertainties in ice phenology estimation partly stem from the uncertainty in the 10-m ice cover maps and the limited temporal resolution of S1, but may also arise from mismatches in spatial scales between the 1-km ice fraction product and point-scale in-situ observations.

Despite the above constraints, our SAR-based record allows for operational mapping of 1-km ice fraction from 10-m ice/water classifications, and quantifying ice phenology over small water bodies. Integrating multiple satellite products holds promise for generating ice fraction and phenology datasets with further enhanced spatial and temporal coverage (Surdu et al., 2015). For instance, combining with the DW product could enable temporally denser ice observations relative to either data

set. In addition, our dataset of small water bodies can be merged with operational products that focus on relatively large lakes,
such as the ESA Lakes_cci (Carrea et al., 2024). The complementary datasets allow for comprehensive assessment of water
bodies across a wide range of sizes. Our algorithm can be applied to additional SAR sensors. For example, adapting our method
to the upcoming NISAR mission could provide independent L-band ice cover observations every 6 days over the globe
(Kellogg et al., 2020). By leveraging multi-source remote sensing of ice dynamics for small water bodies, a more
comprehensive ice fraction and phenology data set can be generated for better monitoring and understanding of the fast-
changing NHL.

## 6 Data availability

The 1-km S1 ice fraction dataset generated in this study, and the code used for its production, are publicly available at
https://doi.org/10.5281/zenodo.17033546 (Lin et al., 2025). The final released dataset is provided in GeoTIFF format with a
spatial resolution of 1 km, a temporal resolution of about 6 days, and is projected in the Alaska Albers Equal Area projection
(EPSG: 3338). Each GeoTIFF image, named as YYYYMMDD.tif, represents the ice fraction of small water bodies in the ACP
on a given day, observed by both ascending and descending Sentinel-1 passes, and recorded as the fraction of ice-covered area
within small water bodies in each 1-km grid cell. The spatial coverage of each product image is consistent with the
corresponding Sentinel-1 acquisition, which may not fully cover the entire study area. Each image contains two bands: (1) ice
fraction, with values ranging from 0 to 1, and (2) the proportion of small water bodies within each 1-km grid cell, also ranging
from 0 to 1. The quality flag information was also provided in the data product in GeoTIFF format, with the band named
"RRMSE" expressed in percentage (%) and representing the RRMSE between S1 and DW ice fraction values for each 1-km
grid cell, calculated using all temporally matched observations over the study period.

## 7 Conclusions

This study used Sentinel-1 SAR imagery, radar backscatter texture features, and air temperature data to develop an ice
fraction dataset for small water bodies (900 m² to 25 km²) across the ACP from 2017 through 2023. The dataset is derived
from 10-m resolution ice cover maps and records the fractional ice cover of small water bodies within each 1-km grid cell in
the ACP, with a temporal resolution of about 6 days. The RF models used for generating the 10-m ice cover maps achieved an
overall accuracy of 0.91, with user and producer accuracies between 0.92 and 0.93. The ice fraction dataset shows strong
agreement with the ice fraction derived from the operational DW product ($R = 0.91$, RMSE = 0.19, RRMSE = 28.41 %, bias
= 0.02). It also yields higher accuracy in estimating ice phenology compared to the DW data (S1 MAE = 7 days; DW MAE =
18 days). Our ice fraction maps show that ice cover in small water bodies across the ACP exhibits high spatial variability
during the thawing period (e.g., June). Ice phenology estimates suggest that ice dynamics of small water bodies in the ACP
are strongly regulated by air temperature, while also being affected by lake and river interactions, and lake properties such as
area and shape. Adapting our algorithm framework to other SAR sensors and integrating other complementary information
from multi-source remote sensing will help improve our products and enable timely monitoring and enhanced understanding
of the changing NHL.

## Author contributions

HL and JD conducted the experiments and drafted the manuscript. JSK and JD supervised the study. All authors contributed
to the manuscript review and revision.

## Competing Interests

The authors declare no competing interests.

## Acknowledgements

This work was conducted at the University of Montana with funding from the National Aeronautics and Space Administration
(80NSSC22K1238).

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
