# Peer review of "A satellite-based ice fraction record for small water bodies of the"

_Earth System Science Data, 2025_

## Author Comment (AC1)

**Response to the reviewer comments**

**Reviewer #1**

The manuscript is a valuable contribution to the delineation of lake ice/water cover using SAR imagery. Although coverage is limited to the ACP, the algorithm shows promise for application to broader areas across the northern hemisphere. The data provides advantages over optical imagery as expected of active microwave. The comparison to DW is a provides suitable validation for the ice fraction product. There are some minor comments that would be good to address. Overall, the manuscript quality is very high but there a few key points that should be addressed.

Thank you for your thorough evaluation and valuable suggestions. We have revised the manuscript accordingly.

There is clear indication that this product could be extended to be an operational product. Was there a reason that other operational products were not compared? For example, the CCI lakes lake ice cover product is available at roughly a 1km resolution and covers some of the lakes in the study area. A comparison to the CCI product would be beneficial due to the similarity between methods, both use a random forest algorithm to classify ice cover.

Thank you for the comment. The Lakes Essential Climate Variable products (Lakes_cci project; https://climate.esa.int/en/projects/lakes/data/) have a spatial resolution of approximately 1 km, and focus on relatively large lakes. Accordingly, the CCI Lakes product has almost no spatial overlap with our data set, which focused on the small water bodies within the ACP study region (Fig. R1).

[Figure]

**Figure R1**

Spatial distributions of small water bodies focused in this study and the lakes in the CCI product (https://climate.esa.int/en/projects/lakes/data/) within the ACP study region.

Considering the complementary nature of the two data sets, we added the following in the Discussion:

*Line 433-435: In addition, our dataset of small water bodies can be merged with operational products that focus on relatively large lakes, such as the ESA Lakes_cci (Carrea et al., 2024). The complementary datasets allow for comprehensive assessment of water bodies across a wide range of sizes.*

**Added reference:**

Carrea, L., Crétaux, J.-F., Liu, X., Wu, Y., Bergé-Nguyen, M., Calmettes, B., Duguay, C., Jiang, D., Merchant, C. J., Mueller, D., Selmes, N., Simis, S., Spyrakos, E., Stelzer, K., Warren, M., Yesou, H., and Zhang, D.: ESA Lakes Climate Change Initiative (Lakes_cci): Lake products, Version 2.1, https://doi.org/10.5285/7FC9DF8070D34CACAB8092E45EF276F1, 2024.

Another question for the authors relates to the choice of texture as a variable for the classifier. The citation provided was conducted for sea ice, however, to the reviewers knowledge no formal exploration of texture has been done for lake ice. Did the authors conduct any investigation into texture values for lake ice? For example, does the texture provide any context for heterogenous surfaces during freeze-up? break-up? Was an investigation done into the temporal evolution of the texture pattern?

As suggested, we examined texture patterns and used the VV correlation texture as an example to illustrate the temporal evolution of texture during the ice break-up period (Fig. S1 in the Supplementary Materials).

For the selected area, the VV correlation texture exhibited a noisy spatial pattern at the onset of ice melt (first row of Fig. S1). The irregular spatial correlations among adjacent pixels indicated heterogeneous snow and ice conditions, likely associated with variations in surface roughness, snow wetness, and ice thickness during this transitional phase. As melting progressed, elevated correlation values emerged along the ice-water boundary (second row of Fig. S1), reflecting spatially consistent backscatter from ice slush or saturated, rough ice surfaces within the ice-water transition zone. This pronounced texture signal provided direct spatial evidence for delineating ice-water boundaries. In the late stage of break-up, open water areas exhibited high correlation values, consistent with the relatively homogeneous surface structure of calm water (third row of Fig. S1).

Overall, the VV correlation texture evolved with the changing ice/water conditions and their spatial patterns, thereby providing support for the machine-learning-based classification.

[Figure]

**Figure S1**

The texture derived from SAR imagery provides spatial information for distinguishing lake ice and open water, illustrated here using a selected area within the study region. The figure shows three stages of lake ice break-up: (1) the early stage (first row, 6 June 2022), (2) the rapid melt stage (second row, 20 June 2022), and (3) the late stage (third row, 30 June 2022). For each row, the two panels from left to right represent the Sentinel-2 RGB image, and the correlation texture computed from the VV band.

We added the following clarification in the manuscript:

*Line 201-203: For example, VV_corr texture information is indicative of ice and water conditions (e.g., ice patches, open water patches, and ice-water boundaries) during the break-up period (Fig. S1), and thus supports the machine-learning based classification.*

There is no variable importance analysis provided - was this conducted? It would be of interest to users to see how the valuable the different variables were in the random forest classifier. The classifier used both SAR parameters and temperature variables, how does the model rate these? The concern here being that the classifier is being driven by temperature rather than SAR/EO data which is the original goal.

We conducted analysis on variable importance and provided Fig. S3 in the Supplementary Materials. Both temperature-based and radar-based features are important, with comparable contributions to the predictions (Fig. S3). For example, in the descending-orbit model, Tmax accounts for 17.6% while VV accounts for 16.8%, and in the ascending-orbit model, Tmean5d accounts for 18.2% while VV accounts for 16.4% (Fig. S3). The temperature variables provide regional temperature distributions and seasonal context, whereas the SAR variables provide the direct observations crucial for distinguishing pixel-level ice conditions.

[Figure]

**Figure S3**

Feature importance of the random forest classifiers trained with ascending-pass data (a) and descending-pass data (b).

We added the following in the manuscript:

***Line 289-291:*** *Both temperature-based and radar-based features are important, with comparable contributions to the predictions (Fig. S3). The temperature variables provide regional temperature distributions and seasonal context, whereas the SAR variables provide the direct observations crucial for distinguishing pixel-level ice conditions.*

---

## Author Comment (AC2)

**Response to the reviewer comments**

**Reviewer #2**

This work introduces a new product to support ice cover research in the north. It also presents a novel method that can be used in other regions to monitor ice cover fraction. This will be highly beneficial for those working in northern areas. The writing is clear and well-structured, and the data is easily accessible and well-labelled for the most part. I have only a few minor suggestions to improve clarity for the reader or user.

*Thank you for your thorough evaluation and your valuable suggestions. We have revised the manuscript accordingly.*

'small water bodies', to me, refers to lakes, rather than rivers – so the validation sites being rivers seemed to come out of the blue while reading. The authors should consider clarifying in the abstract that they are referring to lakes and rivers as small water bodies.

*Thank you for the suggestion. We have clarified the scope of "small water bodies" in the revised abstract.*

***Line 15-17:*** *Here, we developed an ice fraction dataset for small water bodies (ponds, lakes and rivers; 900 m² to 25 km²) across the Arctic Coastal Plain of Alaska (ACP) from 2017 through 2023, using Sentinel-1 Synthetic Aperture Radar (SAR) imagery, texture features, and Daymet air temperature data.*

While I understood the product had 2 bands of data, it wasn't clear to me that each image did not cover the entire study area, so some clarification on that could be added to the text.

*Thank you for the suggestion. We have added this clarification in Section 6 (Data availability) of the manuscript and in the product's README.md file.*

***README.md file & Manuscript Section 6 (Data availability), Line 446-447:***

*The spatial coverage of each product image is consistent with the corresponding Sentinel-1 acquisition, which may not fully cover the entire study area.*

Overall model performance shows many pixels in the moderate-to-large error category (section starting around line 304). Given the data limitations, I agree that this is still a very useful product. The per-pixel quality product, however, could use some clarification. The text lists the values as percentages, but the product loads with a scale of 0 – 153.8. What is the link between uncertainty and the RRMSE values in the quality file? Perhaps even just in the .md file, some explanation of what exactly the RRMSE in the tif are in terms of uncertainty would be helpful.

Thank you for the comment. In the quality file 'grid_rrmse_quality_layer.tif', the value of each 1-km grid cell represents the RRMSE calculated from all same-day paired S1 and DW ice-fraction observations for that grid cell during the study period. The RRMSE values in 'grid_rrmse_quality_layer.tif' are expressed in percentage (%), so a range of 0–153.8 corresponds to 0%–153.8%.

We have added this clarification in Section 3.7 (Uncertainty assessment), Section 6 (Data availability) and in the README.md file.

***Section 3.7 Uncertainty assessment, Line 276-284:***

*To assess the S1 ice fraction data uncertainty, we collected all paired DW and S1 ice fraction observations on the same dates from 2017 through 2023 for each grid cell. Subsequently, the RMSE of S1 and DW ice fractions was calculated for each grid cell and normalized by the average DW ice fraction of that grid cell across all temporally matched observations, to derive the RRMSE, expressed in percentage. The RRMSE for each grid cell is calculated as follows:*

$$RRMSE = \frac{\sqrt{\frac{1}{n}\sum_{i=1}^{n}(S1_i - DW_i)^2}}{\overline{DW}} \times 100\% \qquad (3)$$

*where $S1_i$ and $DW_i$ denote the S1 and DW ice fraction of the same 1 km grid cell at the $i$-th temporally matched observation, respectively; $n$ is the number of temporally matched observations available for that grid cell; and $\overline{DW}$ represents the mean DW ice fraction of that grid cell across all $n$ temporally matched observations. The resulting RRMSE map serves as a quality flag layer for the ice fraction product and will be released alongside the final dataset (Fig. S2, Section 6).*

***Manuscript Section 6 (Data availability), Line 449-451:** The quality flag information was also provided in the data product in GeoTIFF format, with the band named "RRMSE" expressed in percentage (%) and representing the RRMSE between S1 and DW ice fraction values for each 1-km grid cell, calculated using all temporally matched observations over the study period.*

*__README.md file:__ In the quality file 'grid_rrmse_quality_layer.tif', the value of each 1-km grid cell represents the Relative Root Mean Square Error (RRMSE), expressed in percentage (%), calculated from all same-day paired S1 and DW ice-fraction observations for that grid cell during the study period. Specifically, we collected all paired DW and S1 ice fraction observations on the same dates from 2017 through 2023 for each grid cell. Subsequently, the RMSE of S1 and DW ice fractions was calculated for each grid cell and normalized by the average DW ice fraction of that grid cell across all temporally matched observations, to derive the RRMSE.*

Also, can the user be given some cautions to watch for regarding reasons for high RRMSE? e.g., some of the larger errors occur in these regions (it did not appear to me to be particularly spatially based, from a brief review of the data product), or on this size or type of water body, etc.? or is there no discernable set of reasons? With the understanding that this is a data paper and not the venue for a deep exploration of the reasons, a brief comment or two to help the user would be beneficial.

Thank you for this suggestion. We have added a quick view of the quality layer (Fig. S2) to the Supplementary Materials. We found that high errors primarily occur in areas of braided rivers and smaller lakes and ponds, where mixed land and water/ice conditions are likely found in S1 observations.

Accordingly, the following discussions were added in the revised manuscript:

*__Line 321-325:__ Relatively high errors were mainly found along rivers as well as in very small water bodies, where mixed land and water/ice conditions are likely found in S1 observations (Fig. S2). For example, among the 1-km grid cells with RRMSE greater than 60%, 46.7% areas are distributed in river areas determined by a 1 km buffer around the river centerlines. Contaminations in S1 observations from the surrounding land areas of the elongated or very small water bodies likely led to the large classification uncertainties.*

[Figure]

**Figure S2**

The quality layer of the S1 ice fraction product in this study shows that the areas with larger errors are mainly located along rivers and their surrounding regions, as well as in very small lakes and ponds within the study area. The quality layer provides an evaluation of the ice fraction quality for each 1-km grid cell, where each cell's value represents the Relative Root Mean Square Error (RRMSE) between all same-day Sentinel-1 and Dynamic World ice fraction data pairs during the study period (2017–2023), expressed in percentage (%) .

A few minor things to note:

Figure 1: The map should include a panel with an overview of the site's location for context. Even just an outline of Alaska would help to see where it is situated.

Thanks for the suggestion! We have accordingly added an overview panel in Figure 1 to show the location of the study area (below).

[Figure]

**Figure 1.** Numerous small water bodies are distributed across the study area. a, Small water bodies (blue) investigated within the ACP, locations of observed ice phenology records (green triangles), and the three selected regions for ice

phenology analysis (orange circles). b–d, Enlarged views of the three selected regions, with orange borders indicating 5 × 5 km areas. Panels b–d use basemaps from Esri World Imagery.

Line 193: The authors explain that ascending and descending are processed separately, but don't mention why. For clarity, it might be helpful to mention here for those less familiar with radar and the orbital times. (I fully agree with the methods used and the separate processing; this is just a clarification.)

Thanks for the suggestion. We added the following in the revised manuscript:

*Line 228-230: Considering the different passing time, S1 ascending (6 PM local time) and descending (6 AM local time) observations were processed separately.*

Figure 2: b) Constructing the dataset using the 4 types of input data is clear. In panel C, then, the data set goes through the RF model, and it appears that two of the original datasets are then used again to generate the ice cover maps.    The text makes it clear that the RF model was applied to S1 to generate the 10m map. Perhaps the authors could make panel C clearer for workflow, but this might just be a matter of my interpretation.

Thank you for the comment. Figure 2b involves pairing same-day S1, S2, DW, and Daymet data to create the S1-S2-DW-Daymet dataset. The purpose of this dataset is to collect training and testing sample points. Due to the same-day pairing, the S1-S2-DW-Daymet dataset does not include all S1 images. In Figure 2c, we applied the trained classifier to every S1 image during the study period, and therefore, we did not use the paired S1-S2-DW-Daymet dataset from Figure 2b. We have also adjusted the direction of the arrows between the "Processed S1 SAR data" and "Processed Daymet data" boxes in Figure 2c for clarity. The revised Figure 2 is shown below.

[Figure]

**Figure 2.** Workflow for generating ice fraction dataset, comparing S1 and DW ice fraction, and analyzing ice phenology.

Line 344: "This discrepancy may result from a mismatch between the observed freeze-up phase and the phase captured by remote sensing," can the authors clarify what they mean here?

The S1-based ice fraction data captured the ice phenology within 1-km grid cells, whereas the in-situ data set were from eye-based visual observations. Therefore, the two phenology measurements may differ due to mismatches in spatial extent and time of observation.

We have revised the sentence for clarity:

*Line 358-360: The S1-based ice fraction data captured the ice phenology within 1-km grid cells, whereas the in-situ data set were from eye-based visual observations. Therefore, the two phenology measurements may differ due to mismatches in spatial extent and time of observation.*

---

## Author Response (AR2)

**Response to the Editor's Comments**

Dear Editor,

Thank you very much for your comments. Below please find the point-by-point responses (marked in blue).

1. Please undertake one minor change related to the title. Instead of 'Data and Code for paper "A satellite-based ice fraction record for small water bodies of the Arctic Coastal Plain"' change it to a title without 'paper' to A satellite-based ice fraction record for small water bodies of the Arctic Coastal Plain (2017 to 2023) – Dataset and code and link to the ESSD discussion paper (later to the final ESSD publication).

Thank you for the suggestion. We have revised the dataset title accordingly and added a link to the ESSD manuscript on the Zenodo page. A screenshot of the Zenodo webpage is shown below: https://doi.org/10.5281/zenodo.17033546

2. You could also introduce the time frame 2017 to 2023 in the title of your ESSD paper: A satellite-based ice fraction record for small water bodies of the Arctic Coastal Plain (2017 to 2023)

Thank you for the suggestion. We have revised the manuscript title accordingly.

**The first page of the revised manuscript:**

1 **A satellite-based ice fraction record for small water bodies of the**
2 **Arctic Coastal Plain (2017 to 2023)**

3 Hong Lin[1], Jinyang Du[1], John S. Kimball[1], Xiao Cheng[2], J. Patrick Donnelly[1,3], Jennifer D. Watts[4], Annett
4 Bartsch[5]

5 [1] Numerical Terradynamic Simulation Group, University of Montana, Missoula MT, USA
6 [2] School of Geospatial Engineering and Science, Sun Yat-sen University, and Southern Marine Science and Engineering
7 Guangdong Laboratory (Zhuhai), Zhuhai 519082, China
8 [3] Ducks Unlimited Inc., Missoula MT, USA
9 [4] Woodwell Climate Research Center, Falmouth, MA, 02540, USA
10 [5] b.geos, Industriestrasse 1, 2100 Korneuburg, Austria
11 *Correspondence to*: Jinyang Du (jinyang.du@ntsg.umt.edu) and Xiao Cheng (chengxiao9@mail.sysu.edu.cn)

**The first page of the revised Supplementary Materials:**

Supplementary Materials for

**A satellite-based ice fraction record for small water bodies of the Arctic**
**Coastal Plain (2017 to 2023)**

Hong Lin[1], Jinyang Du[1], John S. Kimball[1], Xiao Cheng[2], J. Patrick Donnelly[1,3],
Jennifer D. Watts[4], Annett Bartsch[5]

[1] Numerical Terradynamic Simulation Group, University of Montana, Missoula MT, USA
[2] School of Geospatial Engineering and Science, Sun Yat-sen University, and Southern Marine
Science and Engineering Guangdong Laboratory (Zhuhai), Zhuhai 519082, China
[3] Ducks Unlimited Inc., Missoula MT, USA
[4] Woodwell Climate Research Center, Falmouth, MA, 02540, USA
[5] b.geos, Industriestrasse 1, 2100 Korneuburg, Austria
*Correspondence to*: jinyang.du@ntsg.umt.edu; chengxiao9@mail.sysu.edu.cn